# Probing into the In-Situ Exsolution Mechanism of Metal Nanoparticles from Doped Ceria Host

**DOI:** 10.3390/nano11082114

**Published:** 2021-08-19

**Authors:** Lifang Zhang, Weiwei Ji, Qiyang Guo, Yu Cheng, Xiaojuan Liu, Hongbin Lu, Hong Dai

**Affiliations:** 1School of Chemistry and Chemical Engineering, Nantong University, Nantong 226009, China; lfzhang@ntu.edu.cn (L.Z.); 1808041069@stmail.ntu.edu.cn (W.J.); gyguo@ntu.edu.cn (Q.G.); 2State Key Laboratory of Rare Earth Resource Utilization, Changchun Institute of Applied Chemistry, Chinese Academy of Sciences, Changchun 130022, China; lxjuan@caic.ac.cn; 3University of Science and Technology of China, Hefei 230026, China

**Keywords:** in-situ exsolution mechanism, doped ceria, metal nanoparticle

## Abstract

Exsolved nanoparticle catalysts have recently attracted broad research interest as they simultaneously combine the features of catalytic activity and chemical stability in various applications of energy conversion and storage. As the internal mechanism of in-situ exsolution is of prime significance for the optimization of its strategy, comprehensive research focused on the behaviors of in-situ segregation for metal (Mn, Fe, Co, Ni, Cu, Ag, Pt and Au)-substituted CeO_2_ is reported using first-principles calculations. An interesting link between the behaviors of metal growth from the ceria host and their microelectronic reconfigurations was established to understand the inherent attribute of metal self-regeneration, where a stair-stepping charge difference served as the inner driving force existing along the exsolving pathway, and the weak metal-coordinate associations synergistically facilitate the ceria’s in-situ growth. We hope that these new insights provide a microscopic insight into the physics of in-situ exsolution to gain a guideline for the design of nanoparticle socketed catalysts from bottom to top.

## 1. Introduction

Metal nanoparticle-tailored surfaces with high catalytic active-site density exhibit excellent electrocatalytic and/or photocatalytic performances in energy storages and converters such as storage batteries, fuel cells and photocatalytic water splitting [1,2,3,4,5]. Many tremendous efforts, for example, the deposition techniques through chemical or physical methods, have been devoted to design and construct the metal nanoparticles distributing onto the surface as the active sites during the catalysis [6,7,8]. However, the nanocatalyst obtained from the conventional impregnation approaches often suffers not only the uncontrollability of nanoparticles (size, distribution ratio and anchor sites) during preparation, but also the instability of functional surface due to nanoparticles agglomeration in the working environment, especially for the high-temperature service such as the traditional solid oxide fuel cells (SOFCs) and solid oxide electrolysis cells (SOECs) [8,9]. Therefore, in-situ exsolution, in which nanoparticles directly grow from the parent material and strongly anchor at its surface by the applications of reduction, thermal treatment or electrochemical potential, has received rising attention [10,11,12,13].

Because of its facile process without expensive precursors and multiple “impregnation” steps, in-situ exsolution has become one of the promising, time- and cost-effective strategies to manufacture the controllable and stable nano-catalysts. Metal nanoparticles, such as precious metals (Ag, Pt, Au, etc.) or transition metals (Ni, Co, Cu, etc.), have been successfully exsolved from ABO_3_-type perovskites [14,15,16,17,18,19]. Taking LaFe_1−*x*_M*_x_*O_3_ (M represents the metals elements) as an example, the catalytically active metal ions M are used as the dopants assimilated into the B-site of parent material LaFeO_3_, which would subsequently in-situ socket onto the surface forming metal nanoparticles under the reducing atmosphere [20,21,22]. As compared to the traditional deposition techniques, the in-situ growth of nanoparticles “pinned” on the perovskite surfaces exhibit excellent electrochemical activities and exceptional stabilities in automotive emission catalysts, as well as the novel electrodes for SOFCs and SOECs [23,24,25]. According to a great deal of studies, it is largely accepted that there are two core requirements in an in-situ exsolution process: one is the solubility for these catalyst elements in parent-oxides under the oxidizing conditions, and another is that the formation energies of nanoparticle-decorated surface must be low enough for the catalyst elements’ egress and anchorage [26,27,28]. Obviously, the influence factors contributing to the quality of in-situ exsolving nanoparticles are very complicated, which include surface orientation and composition, catalyst element, deficiency species, reductive condition and so on. For example, a previous article indicated that the expected Ni-nanoparticles were successfully formulated in the La_0.4_Sr_0.4_Ti_0.97_Ni_0.03_O_31−δ_ system as an increased concentration of surface La-ion [29]. Irvine group showed that the A-site deficiency in the AB_11−*x*_M_x_O_3−δ_ accelerates the in-situ growth of transition metal nanoparticles [30]. Recently, Gao et al. revealed that Ni preferentially segregates toward the 100-oriented and SrTiO-terminated surface with ab initio computations, which was consistent with previous experiments [31].

Over the last few decades, the most frequent studies for the in-situ growth phenomenon have focused on the material families of perovskite oxides. With the exception of the perovskite families, the strategy of in-situ exsolution branches out to the rutile, layered K_2_NiF_4_-type, fluorite-structured oxides, etc., which also possess favorable electrochemical performance as electrodes [32,33,34,35]. Boulfrad et al. applied the technique of in-situ exsolution to the rutile-type structure NbTi_0.5_Ni_0.5_O_4_ (NTNO) and showed the potential of nano-scaled Ni particles decorated NTNO as a SOFC anode [36]. Sengodan et al. fabricated layered perovskite PrBaMn_2_O_5+δ_ (PBMO) by in-situ annealing of Pr_0.5_Ba_0.5_MnO_3−δ_, which exhibited high electrical conductivity and remarkable redox stability as ceramic anode material for SOFC [37]. Moreover, Pilger et al. presented a possible segregation strategy for the fabrication of size controllable Pt particles on fluorite-structured ceria via an incorporation−segregation mechanism [38]. It was found that Pt/CeO_2_ nanomaterial with the segregated Pt particles had a nature of remarkable activity and thermal stability as a catalyst in CO oxidation. However, up to now, a systematic regulation of in-situ growth is still unavailable for the exsolution phenomena, such as the external condition in the process of in-situ segregation, the inner driving force for nanoparticle emergence, as well as the identifiable characteristic of parent materials, especially from micro-perspective.

As is well-known, fluorite-structured CeO_2_ is widely utilized in various catalytic processes not only as a support or additive, but also as an active phase of the reaction, particularly in a SOFC system, due to its excellent performance in intermediate/low-temperature conditions [39,40,41]. Implementing the strategy of in-situ exsolution to metal-doped ceria, their catalytic activity and thermal stability can be further improved as reported previously. For instance, more recently, Tan et al. demonstrated that Ni nanoparticles could be strongly attached to the surface of Ni:Gd co-doped CeO_2_ (NGDC) by exsolution procedures [42]. The Ni exsolved NGDC as the anode functional layer revealed a great promotion on the electrochemical performance at the anode supported SOFC under the lower temperature condition. In principle, it is of vital importance to recognize the determining factors of nanoparticle growth, which could make it possible to precisely control the qualities of nanoparticles, so that enhancing the catalytic activity of nanocatalysts with a few strategy optimizations. Nevertheless, there is no in-depth understanding of metal exsolution from the ceria host, and the studies concerning their mechanism of in-situ exsolution are rarely reported.

Herein, taking CeO_2_ as the parent material, we focus on the behaviors of in-situ segregation in M*_x_*Ce_1−*x*_O_2_ (M including transition metals: Mn, Fe, Co, Ni and Cu, and precious metals: Ag, Pt and Au) by using density functional theory (DFT) simulations, aiming to establish a comprehensive understanding about the strategy of in-situ exsolution. In particular, the influencing factors—species of exsolved elements, three different orientation surfaces (001)-, (011)- and (111)-oriented, respectively, and their microstructures—are detailed investigated to find out the key determinant in the CeO_2_ host system. More crucially, the relationship between in-situ exsolution behavior and electronic structures is systematically analyzed to illustrate the inner driving force of nanoparticle exsolution. The analyses of other influence factors, for example, oxygen vacancy, will be the subject of our future work.

### Computational Details

All of our first-principles calculations were implemented in the Vienna ab-initio simulation package (VASP) with the projector-augmented wave (PAW) approach. The generalized gradient approximation (GGA) with the basic of the modified Perdew–Burke–Ernzerhof (PBE_sol) functional was exploited to describe the exchange–correlation interaction [43,44,45,46]. Fluorite-structured CeO_2_ with lattice parameter (a) of 5.411 Å, in accordance with experiment obtained crystal structure [47], was severed as the basic structure to build the investigated models. The kinetic energy cutoff was chosen to be 450 eV. As shown in Figure 1, three kinds of slab models were simulated as the corresponding surface systems. All geometry optimizations were performed within some common criteria, in which ionic relaxation was looped until Hellmann–Feynman force of each atom smaller than 0.005 eV/Å and electronic self-consistent convergence was relaxed to 10^−5^ eV. The spin-polarized method was applied to simulate energetic and electronic structures of each slab model based on the Monkhorst–Pack scheme [48] of 7 × 7 × 7 k-point grid for the 12-atom primitive cell and 3 × 3 × 1 k-point grid for all surface structures with (100)-, (110)- and (111)-oriented. Additionally, the strong correlations inside the metal d electrons and Ce-4f electrons were taken into consideration by the correction of onsite Coulomb interaction U within the setting of U_eff@Ce_ = 5.5 eV, U_eff@M_ = 4.5 eV during all of the DFT simulations, which approximated the previous literature [40,49]. To explore their charge distribution, the Bader method was used to estimate the effective charges of each atom availably [50], and the visualization for electronic and structural analysis (VESTA) was utilized to visualize the charge density distribution both from two-dimensional and three-dimensional perspectives straightforwardly [51].

## 2. Results and Discussion

### 2.1. Construction of Metal Exsolved Surface Structures M_x_Ce_1−x_O_2_

To find out the factors of in-situ exsolution, taking the cubic-fluorite CeO_2_ as the parent material [52], three types of slab models with different orientations were fabricated as the corresponding surface systems. From the side and top views, the (100)-oriented and (110)-oriented CeO_2_ surfaces are comprised of five layers of Ce atoms with 2 Ce in each layer which is labeled by S100 and S110 as shown in Figure 1a,b, respectively. The (111)-oriented model is built of six layers of Ce atoms (Figure 1c) with the sixth layer of Ce and their coordinated O frozen according to previous study [53]. Moreover, each slab model contains a 15 Å vacuum layer which is as far away as possible to avoid interactions between the periodic slabs. As schemed in Figure 1, the upper three layers of all slabs (from the top down) are straightforwardly labeled with Layer 1 (L1 defined as the surface layer), Layer 2 (L2), and Layer 3 (L3 as the bulk layer). Because of their diverse expressions for different kinds of metal nanoparticles from the in-situ exsolution strategy, a series of metal atoms (M), including transition metal (Mn, Fe, Co, Ni and Cu) and precious metal (Ag, Pt, and Au), are taken into consideration to explore exsolution process in terms of element species [4,39]. Therefore, according to the previous literature, the Ce atom located at different layers was comparatively substituted with the M atom to simulate the process of nanoparticle segregation. Here, M substitution in L3 is defined as the solid-solution state, which means M is solved in the parent material, and M at L1 represents the surface segregated states to fabricate the nanoparticles anchored structures [27,31].

It is noticeable that surface stability is the prerequisite for the nanoparticle exsolution system, which can be evaluated by the DFT calculations of the surface formation energy ($E_{f}^{\mathrm{surf}}$) given by the formula as following [54]:(1)$$E_{f}^{\mathrm{surf}}=\frac{1}{2S}\left(E_{\mathrm{slab}}^{M_{x}{\mathrm{Ce}}_{1-x}O_{2}}-nE_{\mathrm{bulk}}^{{\mathrm{CeO}}_{2}}-mE_{M}+mE_{\mathrm{Ce}}\right)$$
where $E_{\mathrm{slab}}^{M_{x}{\mathrm{Ce}}_{1-x}O_{2}}$ is the total energy of slab containing an M substitution M_*x*_Ce_1−*x*_O_2_, $E_{\mathrm{bulk}}^{{\mathrm{CeO}}_{2}}$ is the total energy of pristine CeO_2_ as the bulk phase, *E*_Ce_ and *E*_M_ are the chemical potential of the bulk phase for Ce and substituted atom M. *n* is the stoichiometric coefficient which equals the size of the surface slab, m is the number of the substituted atom, *S* is the surface area (with the unit of Å^2^). The evolutions of $E_{f}^{\mathrm{surf}}$ as a function of M substitutions are obtained as summarized in Figure 2a for the surface of S100, Figure 2b for S110 and Figure 2c for S111. As we know, the negative value of *E*_surf_ suggests that the surface sample is energetically favorable, while the positive value means the instability of the surface system. Therefore, the undecorated CeO_2_ surfaces S100, S110 and S111 with the positive values of $E_{f}^{\mathrm{surf}}$ are identified as the energetically unfavorable samples, which are in the order of S100 > S110 > S111. Obviously, S100 without M substitution is the most unstable surface to a certain extent. It is widely accepted that Ce cations are cubically (8-fold) coordinated with high symmetry in pristine CeO_2_, while in CeO_2_ surface, the exposed terminals with the unfavorable configuration but without the charge neutralization may theoretically cause their thermodynamic instability.

As the previous studies stated, aliovalent doping in the bulk phase of CeO_2_ would give rise to instability, forming an oxygen vacancy or other structural defects in the crystal structures finally [40,55]. However, in this surface research, it is found that all of the investigated M substitutions can stabilize the surfaces in reverse. The substituting process may cause the charge neutralization to balance the surface structure through the inducement of symmetry breaking. As shown in Figure 2, the negative values, closely related to the type of substituted atoms, indicate the potentiality of nanoparticle anchored surfaces. Generally, as a comprehensive comparison between S100, S110 and S111 indicated, the system of M substituted S100 is the most stable among the three types of surface orientations; S111 follows closely and S110 comes last, which seems opposite to the undecorated cases. Moreover, it should be noted that in the transition metal substituted systems, Mn, Fe, Co Ni and Cu are favorably located at the ceria surfaces and their stability increases orderly from Mn to Cu. Meanwhile, for precious metal cases, the abilities of surface formation for Ag and Au cases are energetically similar and the Pt case falls behind slightly. Although the difference among the M-doped systems of S100, S110 and S111 is conspicuous, their evolution tendencies along with Mn → Fe → Co → Ni → Cu → Ag → Pt → Au seem to be consistent apparently. The result outlined above shows that Cu is the most favorable case to locate on the surface of S100, and the stability of the M-anchored surface is strongly affected by the surface orientation, as well as the variety of M substitution.

From the viewpoint of surface stability, a brief hypothesis that the series of doped M can strongly bound to the surfaces of CeO_2_ in the relevant orientations is given according to the investigation of surface energy, which is approximately consistent with the previous experiments [38,42,56]. However, it only proves the stability of M-substituted surface systems presumably attributed to the surface rearrangements with local distortions. There is no direct evidence to verify the feasibility of the process of M in-situ exsolution. Therefore, in the following, we directly estimate the surface segregation energy to study the energetics of the M self-regeneration process, and furtherly reveal the inner mechanism of in-situ exsolution, which has an association with the microscopic electronic configurations of surfaces.

### 2.2. Capabilities of In-Situ Exsolution in M_x_Ce_1−x_O_2_ Systems

As is well-known that it is hard to accurately observe the in-situ exsolution of experimental measurements. Taking the advantage of the DFT method which could fabricate some theoretical structures on the basis of idealized speculations, we first construct the series of M*_x_*Ce_1−*x*_O_2_ (see surface models in Figure 1a–c) to simulate the self-regeneration process of M from the ceria host. Figure 3a depicts a schematic diagram for the process of M segregation, possessing some similarities with the previous theoretical studies. As clarified above, the slab model with M substitution in L1 is regarded as the surface segregated state, and its energy is $E_{\mathrm{surf}}^{M}$, while the slab with M in L3 is defined as the solid-solution state corresponding to the M in ceria bulk whose energy is $E_{\mathrm{bulk}}^{M}$. Based on these definitions, we defined the energy difference between the system with M at L1 and the corresponding one with M in L3 as M segregation energy ($E_{\mathrm{seg}}^{M}$) [27,31]:(2)$$E_{\mathrm{seg}}^{M}=E_{\mathrm{surf}}^{M}-E_{\mathrm{bulk}}^{M}$$

It is widely accepted that $E_{\mathrm{seg}}^{M}$ is a direct energy-descriptor to probe into the feasibility of the M exsolution. As previous research suggested, when $E_{\mathrm{seg}}^{M}$ > 0, the doped M would be energetically stable in the bulk, while if $E_{\mathrm{seg}}^{M}$ < 0, M is preferably exsolved from the lattice to form an M nanoparticle protruded onto the surface. The calculation assessment of $E_{\mathrm{seg}}^{M}$ for each M substituted slab is summarized in Figure 3b, which shows the abilities of M to take part in the in-situ exsolution. Generally, all slab models expect for Co*_x_*Ce_1−*x*_O_2_-S110 case display $E_{\mathrm{seg}}^{M}$ < 0. With the comparison of $E_{\mathrm{seg}}^{M}$ between M-doped S100, S110 and S111, it is worth noting that S100 has the smallest $E_{\mathrm{seg}}^{M}$. Obviously, the big differences of $E_{\mathrm{seg}}^{M}$ between these three surface systems indicate the strong orientation tendencies of the exsolution in the ceria host and M segregations are prone to anchor on S100, which is consistent with the speculation from the analysis of surface stabilities. In detail, each M substitution has its own orientation selection. Taking Au*_x_*Ce_1−*x*_O_2_ and Co*_x_*Ce_1−*x*_O_2_ cases as examples, the Au-doped case has the smallest $E_{\mathrm{seg}}^{M}$ = −10.964 eV in S100, while only −1.853 eV and −1.137 eV of $E_{\mathrm{seg}}^{M}$ for S110 and S111. The exsolution sequence for Au*_x_*Ce_1−*x*_O_2_ is S100 > S110 > S111, and most of the Au ingredient in the ceria host will selectively exsolve out to anchor on S100. For the Co*_x_*Ce_1−*x*_O_2_ case, there is also a lower negative value of $E_{\mathrm{seg}}^{M}$ for S100 ($E_{\mathrm{seg}}^{M}$ = −7.079 eV), and −1.638 eV of $E_{\mathrm{seg}}^{M}$ for S111. However, the positive value of $E_{\mathrm{seg}}^{M}$ (0.209) clearly points out that Co-nanoparticle is unfavored to exsolve out of the (110)-oriented surface. Co selectively segregates oriented in the order with S100 > S111 > S110. Moreover, we can also see that in the S100 system, all of the precious-metal-doped cases (Au, Pt, and Ag) and part of transition metals (Fe and Co) with comparatively low segregation energy of $E_{\mathrm{seg}}^{M}$ < −5 eV may lead to the acceleration for M exsolution. Thus, the potentialities of M exsolutions are arranged in rank order with Au > Pt > Ag for precious metals and Fe > Co > Cu > Mn > Ni for transition metals.

Additionally, it was reported that the reductive condition of in-situ exsolution made a profound impact on the performance of nanoparticles, which may be relative to the energy barrier of self-regeneration. As schemed above, a “two-step” mechanism of M segregation is assumed in Figure 3a. The first step is M “activization”, in which M rises up from L3 (“bulk”) to L2, the next step is M “stabilization” for M grows up from L2 to L1 (“surface”) [27]. Herein, the relative energy of each exsolution case, which is the energy of each individual M*_x_*Ce_1−*x*_O_2_ slab minus $E_{\mathrm{bulk}}^{M}$, is also estimated to measure the process of M self-regeneration. The energy gap between each step is defined as the limiting energy barrier. Analyzed the variations of the relative energies between three orientation slabs as shown in Figure 3c–e, it can be found that the energy gaps among the two steps in the S100 system are distinct as a whole, which means the pathway of M exsolution along the (100)-orientation is energetically unobstructed. Meanwhile, in the S111 system, even though $E_{\mathrm{seg}}^{M}$ < 0, the gaps are ill-defined at the second step, leading to an impediment for M growth in this step. Thus, from the viewpoint of evolved relative energy, a certain energy gap among each step is expected to play an important role in the efficiency of M self-regeneration.

The above results, taken together, show the segregation ability of each M*_x_*Ce_1−*x*_O_2_, both for transition metals and precious metals, and firstly propose that the related M-substitutions are extraordinarily favored to segregate toward the (100)-oriented surface of ceria host. It is inferred the property of M self-regeneration is closely related to certain local distortions because of the surface rearrangements. In this ceria host, structural distortion certainly has an influence on the M-emergence in some form. In Figure 4, the variations of M*_n_*–O bond length (M*_n_* means M in the *n*th layer) reflect the local structural deformations as the function of M substitutions. On the one hand, the M*_x_*Ce_1−*x*_O_2_-S100 system, which possesses a high degree of variation for the bond length of M_1_–O (Figure 4a) and M_2_-O (Figure 4b) and few changes of M_3_-O (Figure 4c) along the M dopants, has comparatively lower segregation energies in M exsolutions. On the other hand, M*_x_*Ce_1−*x*_O_2_-S110 with small changes in the degree of M*_n_*–O (Figure 4d–f) and M*_x_*Ce_1−*x*_O_2_-S111 with large variation extents of M*_n_*–O (Figure 4g–i) is energetically inefficient in the in-situ growth of M nanoparticles. This means that the degree of local distortion induced by the M segregation is proposed as a key structural parameter on the description of the exsolution feasibility, which is ultimately correlated with the electronic reconstructions derived from the surface rearrangements. Therefore, to relate these above results to relevant microscopic structural reconstruction, we further inspect the electronic properties in the following.

### 2.3. Intrinsic Mechanism of In-Situ Exsolution: Stair-Stepping Charge Difference

To investigate why the behavior of in-situ exsolution is quite different from the surface orientations, we firstly gain a deep insight into the charge density difference distribution of each oriented surface without M substitution, which are shown in Figure 5a (CeO_2_-S100), Figure 5b (CeO_2_-S110) and Figure 5c (CeO_2_-S111). Here, flower-like electron clouds with blue color denote the positive electron density, and the negative electron density is expressed in yellow [57]. For the benefit of expression, the effective charges of relevant Ce-ions are indicated via Bader charge analysis obtained with DFT + *U*. See Figure 5a, it can be noted that the cloud shape of Ce_3_ (with charge of +2.32*e*) and Ce_2_ (+2.25*e*) looks like a “lotus”, which hybridizes with O-2*p* adequately, while the low hybridization between the slender-shaped charge cloud of surface-Ce_1_ (+1.54*e*) and O-2*p* contributes to the instability of surface [58]. Interestingly, a recognizable stair-stepping charge difference is found in CeO_2_-S100, as the effective charge of Ce-ions declined from the bottom (L3) to the top (L1), which provides a facile way for M in-situ segregation. On the other hand, the plots of charge density difference for S110 (Figure 5b) and S111 (Figure 5c) show the high degree of bonding hybridization between Ce_n_ (Ce_3_ 2.36*e*, Ce_2_ 2.36*e* and Ce_1_ 2.27*e* for S110, Ce_3_ 2.38*e*, Ce_2_ 2.43*e* and Ce_1_ 2.38*e* for S111) and corresponding O-2*p*. Their indistinct differences of effective charge may narrow the way of M exsolution. Consequently, it can be put forward that the charge distribution of initial configuration decisively determines the direction of in-situ exsolution, and a stair-stepping charge difference as the inner driving force found in CeO_2_-S100 is beneficial to the M self-regeneration, which is in accordance with the exsolution orientation as calculated above.

Considering the M-doping effect on the potentiality of in-situ exsolution, we also analyze the charge density distribution and effective Bader charge of each M-ion for M-substituted cases. The 3D plots of charge density differences (taking Pt*_x_*Ce_1__−*x*_O_2_ as an example) are shown in Figure 6a (for Pt*_x_*Ce_1__−*x*_O_2_-S100), Appendix A (for Pt*_x_*Ce_1__−*x*_O_2_-S110) and Appendix A (Pt*_x_*Ce_1__−*x*_O_2_-S111), where the dopants (M-ions) including Ce are defined as M_3_ to M_1_ along the pathway of exsolution. For more details, we take Fe*_x_*Ce_1__−*x*_O_2_ as the transition metal example (Appendix A). The remarkable changes of charge density distribution are demonstrated due to the surface rearrangement by M doping. It is clear that a certain degree of electron reconstructions emerges when M steps into the ceria host, which on the one hand compensates for the stabilities of surface slabs with the energy decreasing, and on the other hand bares some low-valent sites with high catalytic activity, especially in the M*_x_*Ce_1__−*x*_O_2_-S100 case. Meanwhile, Figure 6b–d, Appendix A and Appendix A summarize the charge evolutions of metals (from M_1_ to M_3_) for M*_x_*Ce_1__−*x*_O_2_-S100, M*_x_*Ce_1__−*x*_O_2_-S110 and M*_x_*Ce_1__−*x*_O_2_-S111, respectively. It is widely accepted that the low effective charges of metal ions manifest a comparatively weak association with surrounding coordination ions [59]. Here, we can see that the changes in the degree of effective charge reduction for M_1/2/3_ in each layer as a function of M substitutions are approximately consistent with the order of M segregation obtained above. For instance, also taking Pt*_x_*Ce_1__−*x*_O_2_-S100 as the example, the weaker bonding of Pt_1/2/3_-O in Pt*_x_*Ce_1__−*x*_O_2_-S100 because of the low charged Pt-ions leads to a better capability of exsolution. Hence, based on the detailed analysis of Bader charge [60], another determining parameter, the bonding strength between M_1/2/3_-O, is proposed as the second significant factor for in-situ exsolution.

Moreover, for more intuitive information, the sectional drawings of charge density difference along the (001)-direction as shown in Figure 7a and Appendix A (M*_x_*Ce_1__−*x*_O_2_-S100 with Pt/Fe on “surface”), and Figure 7b and Appendix A (M*_x_*Ce_1__−*x*_O_2_-S100 with Pt/Fe in L2) and Figure 7c and Appendix A (M*_x_*Ce_1__−*x*_O_2_-S100 with Pt/Fe in “bulk”), are extracted to unmask the process of in-situ exsolution. In Figure 7c, when Pt is in L3, a characteristic electric potential difference exists along the pathway of Pt migration, which provides an inner drive-force to set the stage for Pt exsolution. When Pt is in L2, as shown in Figure 7b, it is worth noting the Pt-coordinate association—the connection between Pt_2_ and their coordination O ions is rather weak. This infirm association facilitates the Pt’s “escape” from L2 to the surface. Ultimately, the exsolved Pt with the dumbbell-shaped electron cloud (as plotted in Figure 7a) is well anchored on the ceria surface, causing the nano-socketed structure to simultaneously possess surface stability and catalytic activity. To sum up, a reasonable exsolution mechanism for the M*_x_*Ce_1__−*x*_O_2_ surface is proposed in our theoretical study.

## 3. Conclusions

In summary, we theoretically investigate the process of in-situ exsolution for ceria-based fluorite framework M*_x_*Ce_1−*x*_O_2_ (Mn, Fe, Co, Ni, Cu, Ag, Pt and Au). Our calculations of surface formation energy and M segregation energy show that the substituted M preferentially toward the (100)-oriented ceria surfaces and their stabilities and exsolution actives are sensitive to the dopant species M. It is found that Au/Pt/Ag/Fe/Co-doped S100 with smaller segregation energy can energetically accelerate the M self-regeneration, while the exsolution process in Cu/Ni/Mn-doped S100 may need more harsh reduction conditions to urge the M’s ascent. In particular, linking the potentiality of M segregation with their microelectronic structure, our study proposes two crucial synergistic factors to uncover the mechanism for the in-situ exsolution of M-doped ceria: one is the inner driving force, a stair-stepping charge difference existing along the pathway of M migration is considered as the prerequisite factor for M exsolution; another factor is served as “motion plus”, the bonding strength between M_1/2/3_-O, where it is believed that the weak M-coordinate association will accelerate the M’s segregation. As a result, our new insights give a fundamental guideline for the design of the exsolved nanocatalysts, which is a step toward a reasonable understanding of the in-situ growth mechanism.

## Figures and Tables

**Figure 1 nanomaterials-11-02114-f001:**
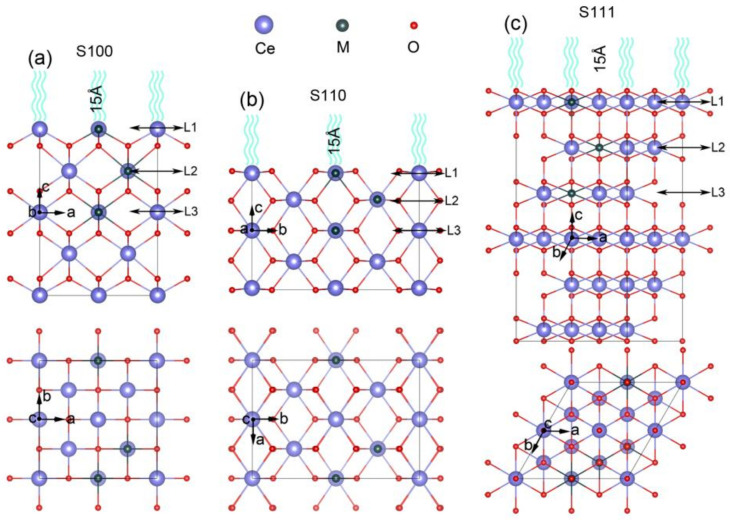
Side views (upper panel) and top views (lower panel) of modeled surfaces for M*_x_*Ce_1−*x*_O_2_. (**a**) 100-oriented slab labeled with S100, (**b**) 110-oriented slab labeled with S110 and (**c**) 111-oriented slab labeled with S111.

**Figure 2 nanomaterials-11-02114-f002:**
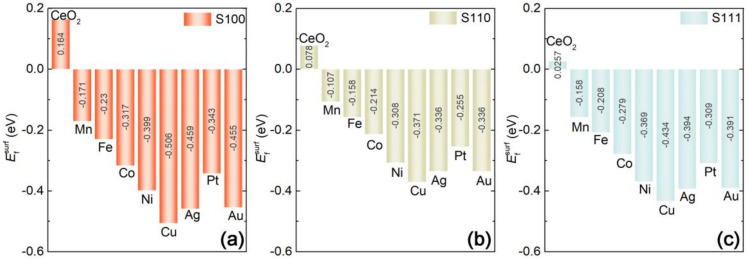
Surfaces formation energy ($E_{f}^{\mathrm{surf}}$) of M*_x_*Ce_1−*x*_O_2_ slabs with different orientations: (**a**) for S100, (**b**) for S110 and (**c**) for S111.

**Figure 3 nanomaterials-11-02114-f003:**
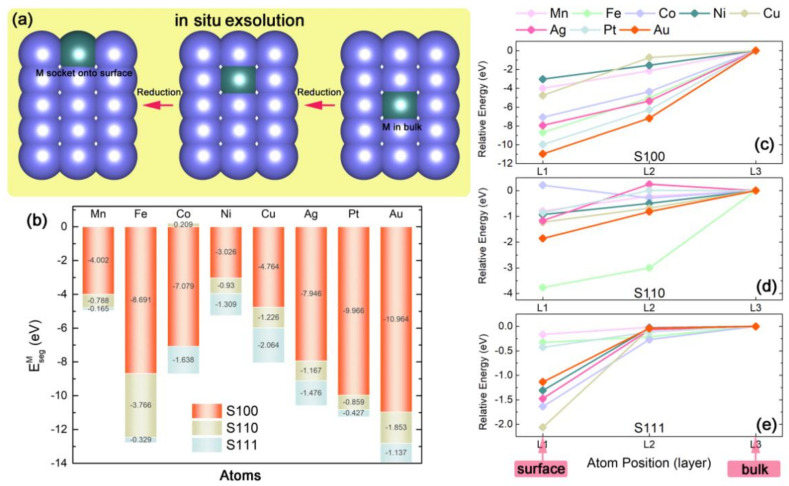
(**a**) The schematic diagram of metal M in-situ exsolution in term of atomic scale. (**b**) Evolution of the segregation energy $E_{\mathrm{seg}}^{M}$ for different oriented M*_x_*Ce_1−*x*_O_2_ slabs as a function of M substitutions. (**c**–**e**) Relative energy of M*_x_*Ce_1−*x*_O_2_ slabs of as a function of the position of metal atoms, (**c**) for S100, (**d**) for S110 and (**e**) for S111, the zero of energy is the energy of M*_x_*Ce_1−*x*_O_2_ slabs with M in the “bulk” as the reference basis.

**Figure 4 nanomaterials-11-02114-f004:**
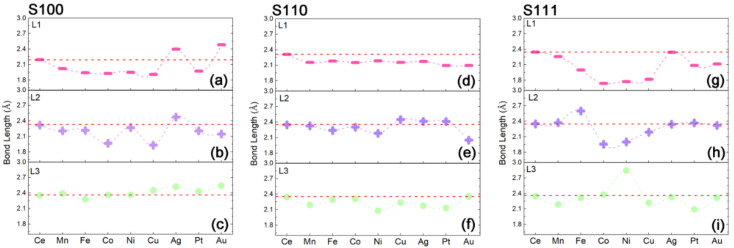
The evolution of M–O bond length as the function of M substitutions for each M*_x_*Ce_1−*x*_O_2_ slab: (**a**–**c**) for S100 system with M in L1 (**a**), L2 (**b**) and L3 (**c**), (**d**–**f**) for S110 system with M in L1 (**d**), L2 (**e**) and L3 (**f**), and (**g**–**i**) for S111 system with M in L1 (**g**), L2 (**h**) and L3 (**i**).

**Figure 5 nanomaterials-11-02114-f005:**
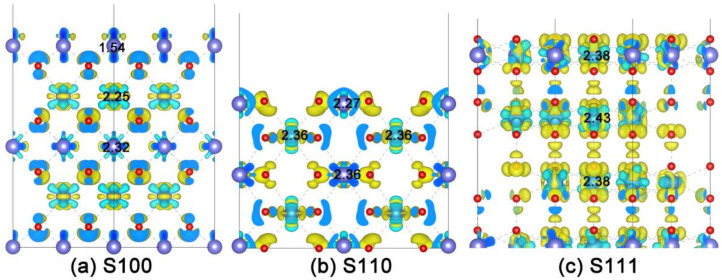
The 3D plots of charge density difference with an isosurface value of 0.02 *e*/Bohr^3^ for pristine CeO_2_ with different oriented surfaces: (**a**) for S100, (**b**) for S110 and (**c**) for S111.

**Figure 6 nanomaterials-11-02114-f006:**
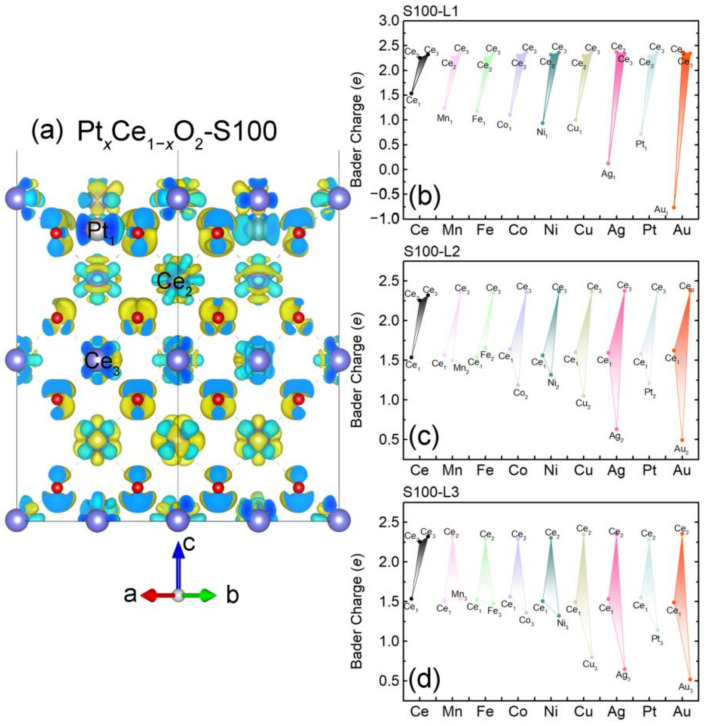
(**a**) The 3D plot of charge density difference with an isosurface value of 0.02 *e*/Bohr^3^ for Pt*_x_*Ce_1−*x*_O_2_-S100 with Pt at L1. (**b**–**d**) Evolution of Bader charge for M*_x_*Ce_1−*x*_O_2_-S100 with the dependence of Mn, Fe, Co, Ni, Cu, Ag, Pt and Au: (**b**) M in L1, (**c**) M in L2 and (**d**) M in L3, the relevant M site are labeled as M_1_, M_2_ and M_3_.

**Figure 7 nanomaterials-11-02114-f007:**
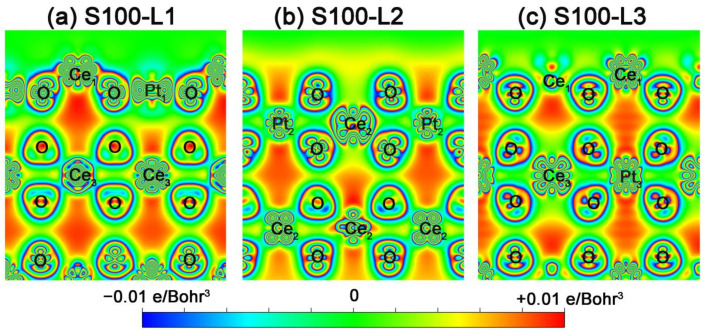
2-D charge density difference plots along the (001)-direction (∆*n* (**r**) in units of *e*/Bohr^3^) for 1.5 × 1.5 × 1 Pt_*x*_Ce_1−*x*_O_2_-S100, which with Pt (**a**) on surface layer L1, (**b**) in second layer L2 and (**c**) in the “bulk” layer L3.

## Data Availability

The data used to support the findings of this study are available subject to approval from the relevant departments through the corresponding author upon request.

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
