# Peer review of "Probing into the In-Situ Exsolution Mechanism of Metal Nanoparticles from Doped Ceria Host"

_nanomaterials, 2021, doi:10.3390/nano11082114_

Round 1

Reviewer 1 Report

This is a nice and well written contribution in the field of catalytic materials, particularly dealing with exsolved nanoparticles based on CeO2 containing eight different transition metals. The approach, based on DFT calculations, is sound and no major technical flaws are detected. Three different surface orientations are investigated, making the analysis complete, although defects are not considered and will be tackled in future studies. Both formation and segregation are discussed and rationalized on electronic density and energetics basis. The major outcome is that Au/Pt/Ag/Fe/Co-doped S100 is optimal. But the novelty is the attempt of identifying an explanation to exsolution, which provides a thorough rationalization of the phenomenon. Undoubtedly I think that this work is worth of publication in Nanomaterials.

Minor: In Figures 2 and 3 I would invert the orientation of the vertical axis (from largest negative to smallest negative/positive): even if the bars point downwards, I find it odd to represent stabilization with this orientation.

Reviewer 2 Report

The paper titled “Probing into the in-situ Exsolution Mechanism of Metal Nanoparticles from Doped Ceria-host” by Zhang et al. investigated the behaviors of transition metal segregation on CeO2. The manuscript was well written, and they properly performed DFT calculations. However, this manuscript does not include enough novelty and information for the simulation details to be published in Nanomaterials. I think this paper cannot be published in this journal for the following reasons.

  • Why authors used the same U value of 4.5eV for different kinds of transition metals? It should be different.
  • Why authors used different numbers of layers for CeO2 100, 110, and 111 (Fig. 1c)? The thickness of slab model can affect the energetics for dopant segregation. The sentence “All of the modeled CeO2 surfaces comprised of five layers of Ce atoms with 2 Ce in each layer” in lines 142 – 143 should be corrected.
  • Exsolution only occurs under reducing environments. Why didn’t the authors consider about oxygen vacancy effect? Metal-oxygen vacancy co-segregation has been widely reported as a key descriptor to predict exsolution tendency in oxide materials [ Commun. 2017, 8, 15967].
  • Why is the trend of energetics between dopant segregation and surface formation different? The fundamentals between them might be the same as the stability of dopants.
  • In general, dopant segregation tendency follows the periodic table [ Commun. 2017, 8, 15967]. Why is that tendency not observed in this paper?
  • Without any direct relationship between segregation energy and M-O bond length (or charge difference), it is hard to agree that the M-O bond length can be a key descriptor of exsolution. It might be helpful to plot a figure showing the relationship between them.
  • Only DFT results should be strict about proposing a hypothesis.

Round 2

Reviewer 2 Report

U value should be optimized by experimental results, such as bandgap or electronic structure. Please read more DFT papers related to U value optimization. In this case, I think results cannot be changed by the U value. However. it can change the trend in some cases.

Now, I think this manuscript can be published in this journal only after all of the responses to my comments are included in the revised manuscript.

Author Response

Your response makes us rejoice and provides us an opportunity to go further on this work. We have reviewed many previous studies and it is indeed that the U value is important in the some cases. And according our test of U values, the results may be insensitive to the U value.  And thanks again for your reviewing and all comments which help us to improve this work a lot!